



# 1 A deep learning method for convective weather
# 2 forecasting : CNN-BiLSTM-AM (version 1.0)

Jianbin Zhang[1,3], Zhiqiu Gao[1,2], Yubin Li[1], Yuncong Jiang[1]
[1] School of Atmospheric Physics, Nanjing University of Information Science & Technology, Nanjing,
210044, China
[2] State Key Laboratory of Planetary boundary layer Physics and Atmospheric chemistry, Institute of
Atmospheric Physics, Chinese Academy of Sciences, Beijing,100081,China
[3] College of Artificial Intelligence, Henan University, Kaifeng, 475000, China
*Correspondence to:* Dr. Zhiqiu Gao (zgao@nuist.edu.cn) and Dr. Yubin Li (liyubin@nuist.edu.cn)
**Abstract.**This work developed a CNN-BiLSTM-AM model for convective weather forecasting using
deep learning algorithms based on reanalysis and forecast data from the NCEP GFS, the performance of
the model was evaluated. The results show that: (1) Compared to traditional machine learning algorithms,
the CNN-BiLSTM-AM model has the ability to automatically learn deeper nonlinear features of
convective weather. As a result, it exhibits higher forecasting accuracy on the convective weather dataset.
Furthermore, as the forecast lead time increases, the information value provided by this model also
changes. (2) In comparison to subjective forecasts by forecasters, the objective forecasting approach of
the CNN-BiLSTM-AM model demonstrates advantages in metrics such as Probability of Detection
(POD), False Alarm Rate (FAR), Threat Score (TS), and Missing Alarm Rate (MAR). Specifically, the
average TS score for heavy precipitation reaches 0.336, which is a 33.2% improvement compared to the
forecaster's score of 0.252. Moreover, due to the CNN-BiLSTM-AM model's ability to automatically
extract classification features based on a large sample dataset and consider a comprehensive range of
convective parameters, it effectively reduces the FAR. (3) The interpretability study of the machine
learning-based convective weather mechanism reveals that the importance ranking of convective weather
forecasting factors arranged by machine learning methods largely aligns with the subjective
understanding of forecasters. For example, the total precipitable water (PWAT) is identified as a critical
factor for short-term heavy precipitation forecasting, regional factors have significant impacts on
convective weather, and vertical motion at 300 hPa provides dynamic lifting conditions for convection.
This objective analysis of factor ranking not only further confirms the effectiveness of machine learning
in automatically extracting convective weather features but also validates the rationality of the sample
set construction. Overall, the use of the CNN-BiLSTM-AM model in convective weather forecasting

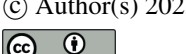



demonstrates superior performance compared to traditional machine learning algorithms and subjective
forecasting methods.
**1 Introduction**
The forecasting of severe convective weather primarily focuses on violent weather phenomena that occur
on small spatial and temporal scales, including hail, thunderstorms, strong winds, short-duration heavy
rainfall, tornadoes, and other hazardous and dangerous weather conditions. Due to the regional intensity
and rapid development characteristics of severe convective weather, forecasting is extremely challenging
(Han et al., 2009; Zheng et al., 2015). Such weather often has a high intensity that can potentially cause
severe casualties and property losses (Wang et al., 2007). For instance, the "Eastern Star" ship sinking
incident in 2015 (Zheng et al., 2016a), the EF4 level tornado event in Funing, Jiangsu in 2016 (Zheng et
al., 2016b), and the massive "720" rainstorm event in Zhengzhou, Henan in 2021 (Gao et al, 2022). In
monitoring and predicting severe convective weather, traditional methods that solely rely on statistical
results or forecaster experience exhibit significant limitations. Firstly, the occurrence condition and
threshold characteristics of severe convective weather will undoubtedly vary among regions with
different seasons, terrains, and climatic backgrounds. This diversity makes it challenging to use a unified
set of physical quantity thresholds to predict categories of severe convective weather in various areas.
Secondly, the volume of data needed to be processed during weather forecasting is considerable, and the
features and threshold ranges of physical quantities extracted by forecasters through statistics or
subjective judgement may not fully capture beneficial information or subtle changes in the data,
especially those at smaller scales. Additionally, due to the complex variations of severe convective
weather, if forecasters lack comprehensive and profound understanding of the rules governing the
development of severe convection, they are unable to completely grasp the useful features during the
development process of such weather. Even though some objective forecasting algorithms exist based
on physical principles and statistical properties of relevant physical quantities, thoroughly extracting
these features remains challenging. Moreover, when extracting features of physical quantities,
forecasters' abilities are constrained by their experiential knowledge and understanding.
The brisk advancement of artificial intelligence in recent years has spurred changes across numerous
domains. Notably, AI algorithms underpinned by deep learning and machine learning have achieved


substantial progress and found effective applications across diverse sectors. The integration of these
algorithms with meteorological big data often forms an efficacious toolset for severe convective weather
forecasting (McGovern et al., 2017; Reichstein et al., 2019). Li et al. (2018) successfully employed the
random forest algorithm to categorize potential severe convective weather phenomena. Based on this,
they developed forecasting models for short-duration intensive rainfall, thunderstorms, hail, and severe
convection. The training of these models involved leveraging convective indices and physical quantities
possessing explicit physical meanings. Furthermore, they integrated real-time forecast field data from
NCEP's Global Forecast System (GFS) into their predictions. The forecast outcomes of 85 severe
convective occurrences indicated a total misjudgment rate of 21.9% with zero omissions, thereby
confirming the model's extensive suitability for severe convective weather predictions. Herman et al.
(2018) employed NOAA's global ensemble forecast system data to construct a machine learning model
by using the random forest algorithm. This model encapsulates not only numerical forecasting elements
like convective indices, temperature, pressure, humidity, and wind field but also includes numerous
background forecasting components such as the maximum, minimum, median, longitude, and latitude
values of 1-year and 10-year average recurrence intervals (ARIs). This model has proven successful in
predicting extreme precipitation events 2-3 days ahead. Using the Bayesian approach, Liu et al. (2019)
conducted correlation analysis for thermal and dynamic factors within high-frequency lightning storm
cloud processes. Their findings underscored convective potential energy, convective inhibitory energy,
and low-level wind shear as the most influential forecasting factors – providing crucial insights for high-
frequency lightning storm cloud predictions. In contrast to traditional machine learning algorithms (e.g.,
shallow neural networks, random forest), deep learning is capable of modeling intricate nonlinear
systems and offering superior levels of abstraction. Importantly, deep learning can express broad function
sets unattainable by shallow networks more flexibly and succinctly. It demonstrates substantial
superiority over traditional methodologies across various fields, including speech processing and image
recognition (Krizhevsky et al., 2012; Lecun et al., 1995; Szegedy et al., 2013). Deep learning has also
found initial applications in short-term meteorological forecasting. For example, Lin et al. (2019) built a
Convolutional Long Short-Term Memory (ConvLSTM) model based on the Weather Research and
Forecasting (WRF) model and lightning data to extract spatiotemporal characteristics for future 12-hour
lightning predictions. Similarly, Gope (2016) constructed a rainstorm prediction model using deep neural





networks (specifically, stacked autoencoders) based on historical climatic data, capable of forecasting
intense rainfall scenarios in regions like Mumbai and Kolkata, India, 6 to 48 hours in advance.
With the continuous improvement of small and medium-scale observation networks, the gradual
enrichment of observational methods, and the rapid growth of observational data, exploring the
occurrence and development mechanisms in each severe convective weather process, identifying
characteristic parameters and threshold ranges of various types of severe convective weather from a large
amount of numerical model data, and comprehensively considering the geographical and climatic
environment of each region have become key factors for effective forecasting of severe convective
weather. Deep learning algorithms can automatically extract important features from big data, deeply
extract effective information, and comprehensively consider geographical and climatic differences across
regions, which will significantly optimize the results of severe convective weather forecasting. The latest
Pangu-Weather model is capable of predicting meteorological elements, including temperature, wind
speed, and pressure, with high accuracy and fast prediction speed (Bi et al., 2023). Based on the
reanalysis and forecast data from NCEP GFS global numerical model, our research uses deep learning
algorithms to establish a severe convective weather forecasting model and objective forecasting method
that can provide real-time objective forecast products nationwide.
**2 Data and methods**
**2.1 NCEP FNL analysis data**
The training and testing data in this study were obtained from the NCEP GFS 0.25°×0.25°FNL (final)
analysis data for the period of 2015-2020. The NCEP FNL analysis data provides global analysis fields
four times daily (02:00, 08:00, 14:00, and 20:00). The selection of forecast factors includes not only
basic meteorological elements such as temperature, geopotential height, humidity, and wind field, but
also commonly used physical quantities reflecting conditions related to water vapor, dynamics, and
energy, such as precipitable water (PWAT), Convective inhibition (CIN), and K-index (Tian et al., 2015).
Additionally, in order to consider geographical variations in convective activity, features such as
elevation, longitude, and latitude are included, resulting in a total of 144 features (see Table 1) to be
analyzed.

115                                    **Table 1. The selection of parameters**





| | parameters | level/(hPa) |
|---|---|---|
| Multi-level physical quantities | T(Temperature)、TD(Dew point) | 1000、925、850、700、500、300、200 |
| | H(Potential height) | |
| | Wind_Speed、Wind_Drection、W(P-coordinate system vertical velocity) | |
| | Rh(Relative humidity)、TDD(Temperature dew point difference)、Q(Specific humidity)、VAPFLUXDIV(Water vapor flux divergence) | |
| | PV(Potential vorticity)、TMPADV(Temperature advection)、SITASE(Pseudo-equivalent potential temperature)、DIV(divergence)、VOR(vorticity)、VORADV(vorticity advection) | |
| Commonly used strong convective physical quantities | BCAPE(Bulk Convective Available Potential Energy)、BLI(Bulk Lifted Index)、CIN(Convective inhibition)、DCAPE(Downdraft Convective Available Potential Energy)、K(K-index)、LI(Lifted Index)、Z0(Altitude of the 0 °temperature layer)、Z20(Altitude of the -20 °temperature layer)、PWAT(Precipitable water)、SHIP(Large hail index)(Allen et al,2015)、SHR1(0-1km wind shear)、SHR3(0-3km wind shear)、SHR6(0-6km wind shear)、SI(Sha index)、TOAT(Total Index) | |
| Other quantities | ELEVATION、LON(Longitude)、LAT(Latitude) | |

**2.2 Severe convective weather data**

In this study, the convective data used were sourced from observational data collected by Chinese surface meteorological stations and archived by the National Meteorological Information Center. The National Meteorological Information Center performs quality checks and error data corrections on timed observations and daily extreme value data for 19 major elements, including temperature, pressure, precipitation, humidity, and sunshine duration, from over 2400 ground stations nationwide. This process

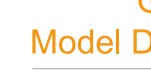 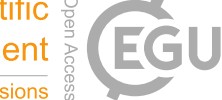

addresses systematic issues that may arise during the digitization process, such as data omissions or
duplication. The data undergoes three levels of quality control review at the station, provincial, and
national levels within the routine operations of the meteorological data departments.
**2.3 Selection of experimental area**
The Henan region is not only one of China's important agricultural production areas but also a zone
where modern cities and rural areas coexist with complex terrain due to extensive industrial development.
Furthermore, being located in the mid-latitudes, Henan frequently experiences cold air invasions, while
warm and humid air masses can also reach the region during the summer, often leading to intense rainfall
events resulting from the convergence of warm and cold air masses. Additionally, numerous signs
indicate that China's climate is entering a transitional period, potentially leading to a shift from low
rainfall in summer to higher rainfall in northern regions. Therefore, selecting the Henan region (as
illustrated in Figure 1) as the study area for this research is highly appropriate. Undoubtedly, it will lay
the foundation for future improvements in severe weather warnings and disaster prevention and
mitigation capabilities in the region, reflecting a forward-looking study with a strategic perspective.

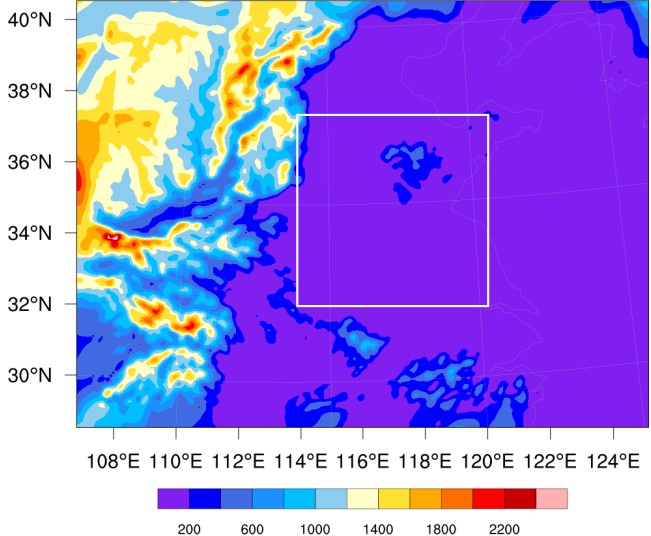


**Figure 1: The selected area of experiment ( The white rectangle denotes the selected area: Henan, 31°-37°**
**N,110°-117° E ); Distribution of topography (shaded; unit: m)**





**3 Deep learning model**

**3.1 Deep learning model structure**

To forecast severe convective weather more accurately, this study develops a CNN-BiLSTM-AM-based

model for convective prediction. The model consists of Convolutional Neural Network (CNN),

Bidirectional Long Short-Term Memory (BiLSTM), and Attention Mechanism (AM). The CNN extracts

features from the input inventory data, while the BiLSTM effectively captures the interdependencies in

the temporal sequence data. The AM is a mechanism that improves results by capturing the impact of

past feature states on heavy rainfall. The model primarily comprises the CNN, BiLSTM, and AM layers,

including an input layer, CNN layer (with one-dimensional convolutional and pooling layers), BiLSTM

layer (with forward and backward LSTM layers), AM layer, and output layer (see Figure 2 for details)(Lu

et al,2021). During weather forecasting, we analyze historical meteorological images and numerical

products to draw conclusions. To summarize this process briefly, we first analyze meteorological images

or various forecast products to generate situational forecasts. Then, combining the situational forecasts

with local real-time meteorological data, we obtain element forecasts. If we simplify these two steps

further, the first step involves extracting features from meteorological images or forecast products (i.e.,

situational forecasting), and the second step involves fitting the extracted features with local historical

meteorological information to obtain the required forecast values (i.e., element forecasting). In the CNN-

BiLSTM-AM model, these two steps are transformed accordingly: using the CNN for data feature

extraction and subsequently applying the BiLSTM and AM to match historical meteorological

information in order to derive the element values. Prior to inputting the data into the CNN-BiLSTM-AM

model, we normalize the data and convert it into matrix form. Once these settings are completed, the

model can be trained.



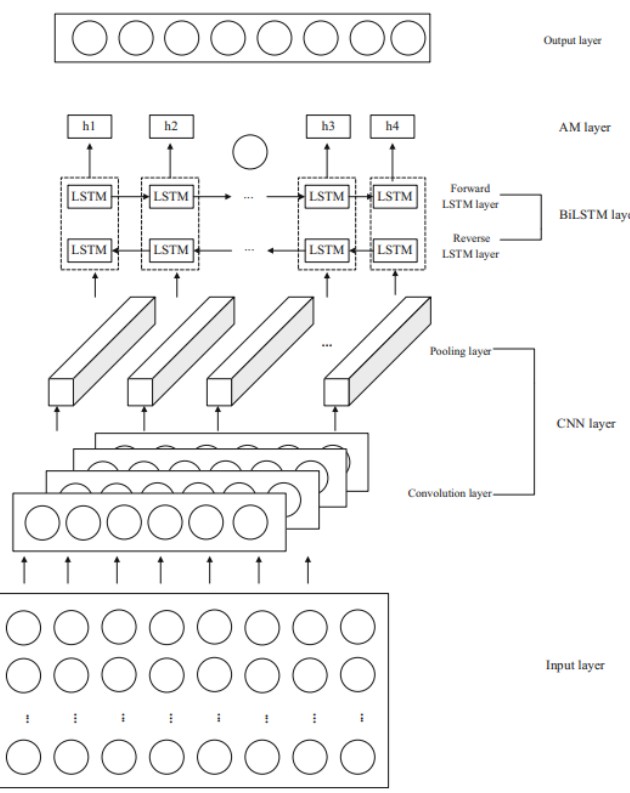


**Figure 2: CNN-BiLSTM-AM model structure diagram**
The training process of CNN-BiLSTM-AM is illustrated in Figure 3, with the main steps outlined as
follows:
1)Input Data: Provide the necessary data for training CNN-BiLSTM-AM.
2)Input Data Normalization: To enhance model training performance, the input data is standardized due
to its significant variations. The normalization formula is expressed as Equation (1):

$$y_i = \frac{x_i - \overline{x}}{s} \qquad (1.1)$$

Among them, $y_i$ is the standardized value, $x_i$ is the input data, $\overline{x}$ is the average value of the input data,
and $s$ is the standard deviation of the input data.
3)Network Initialization: Initialize the weights and biases of each layer in CNN-BiLSTM-AM.
4)CNN Layer Computation: Pass the input data sequentially through the convolutional and pooling
layers within the CNN layer to extract features and obtain output values.





5)BiLSTM Layer Computation: Use the hidden layer of the BiLSTM layer to compute the output data
from the CNN layer and obtain output values.
6)AM Layer Computation: Compute the output data from the BiLSTM layer using the AM layer and
obtain output values.
7)Output Layer Computation: Calculate the output value of the model by computing the output value
of the AM layer.
8)Error Calculation: Compare the computed output value from the output layer with the true value of
the data set and calculate the corresponding error.
9)Check if the termination conditions for the training process are met: Successful termination conditions
include completing a predetermined number of cycles, reaching a weight below a certain threshold, or
achieving a prediction error rate below a specific threshold. If any of these conditions are met, the training
is completed; otherwise, the training continues.
10)Error Backpropagation: Propagate the calculated error in the opposite direction, updating the weights
and biases of each layer, and then return to step 4 to continue the training.



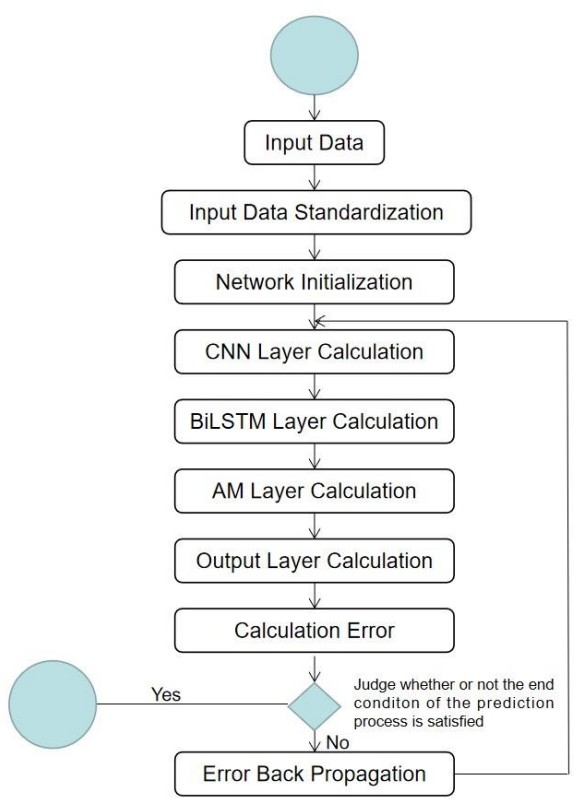

**Figure 3: Flow chart of CNN-BiLSTM-AM training process**
The prediction process of CNN-BiLSTM-AM is illustrated in Figure 4 and consists of the following
main steps:
1) Input Data: Provide the input data required for prediction.
2) Data Standardization: Normalize the input data.
3) Prediction: Feed the standardized data into the trained CNN-BiLSTM-AM model and obtain the
corresponding output values.
4) Data Standardization Recovery: The output values obtained from CNN-BiLSTM-AM are in
standardized form. To restore them to their original values, apply Equation (2) to convert the standardized
values back.

$$x_i = y_i * s + \bar{x} \qquad\qquad (1.2)$$



Among them, $x_i$ represents the recovered value of the standardized value, $y_i$ represents the output
value of CNN-BiLSTM-AM, s represents the standard deviation of the input data, and $\bar{x}$ represents the
mean value of the input data.
5) Output Results: Present the recovered results after restoration as the completion of the prediction
process.

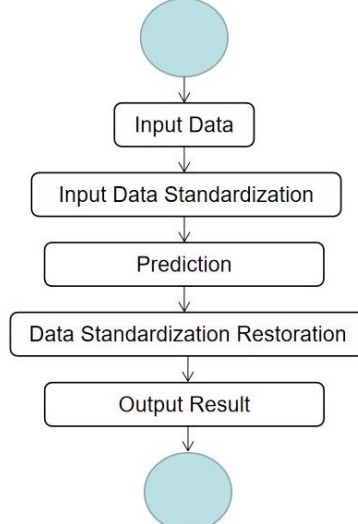


**Figure 4: Flow chart of CNN-BiLSTM-AM prediction process**
**3.2 Construction of training and testing datasets**
The prediction of severe convective weather can be viewed as a binary classification problem,
distinguishing between its occurrence (labeled as 1) and non-occurrence (labeled as 0). Consequently,
we can utilize actual severe weather data to calibrate the numerical analysis field, thus constructing
training and testing sample datasets. Since NCEP FNL provides both analyzed and forecasted grid-based
fields while the observed data is in scattered form, the observed data needs to be gridded. If a related
weather event occurs within a grid radius of R, it is considered that the event occurs at that grid point
(labeled as 1); otherwise, it is considered that the event does not occur at that grid point (labeled as 0).
Considering that convective weather is typically associated with mesoscale weather systems, we
conducted experiments and determined to use R=20km. Setting R too small may result in missing severe
convective events, whereas setting it too large may lead to false alarms.



The occurrence probability of severe convective weather is relatively low, resulting in a significant class
imbalance where positive samples (i.e., samples with severe convective events) are much fewer than
negative samples (i.e., samples without severe convective events). This presents a typical issue of
imbalanced data (Krawczyk et al., 2016). To address this, we employed the oversampling technique
(Buda et al., 2017) by randomly duplicating positive samples to achieve a balanced distribution between
positive and negative samples. With this approach, we constructed the training and testing datasets using
real-time data from January to December between 2015 and 2020, as well as NCEP FNL data. The testing
dataset consisted of one randomly selected day from each month during the aforementioned period,
totaling 72 days, while the training dataset comprised the remaining samples. During the training process,
we utilized the ADAM optimizer (Kingma and Ba, 2014) with a learning rate set at $10^{-4}$, and default
values were used for other settings (Perol et al., 2017). The CNN-BiLSTM-AM model was trained for
30 epochs with a batch size of 64. Through training and parameter tuning, we obtained the optimal
prediction model. This model can incorporate NCEP FNL data and transform features into a four-
dimensional array of M×28×32×1 (M represents the number of samples), enabling predictions for severe
convective weather.
**4 Results**
**4.1 Evaluation methods**
Commonly used verification measures for evaluating forecast results include the Probability of Detection
(POD), the Threat Score (TS), the Equitable Threat Score (ETS), the Bias (BIAS), the False Alarm Ratio
(FAR), and the Missed Alarm Ratio (MAR). These measures are defined as follows:

$$POD = \frac{h}{h + m} \tag{1.3}$$

$$TS = \frac{h}{h + m + f} \tag{1.4}$$

$$ETS = \frac{h - h_{random}}{h + f + m + -h_{random}}, h_{random} = (h + f) * (h + m)/(h + m + f + c) \tag{1.5}$$

$$BIAS = \frac{h + f}{h + m} \tag{1.6}$$





$$FAR = \frac{f}{h + f} \tag{1.7}$$

$$MAR = \frac{m}{h + m} \tag{1.8}$$

Among them, h represents the number of occurrences when both the forecast and observed events appear,
m represents the number of occurrences when the observed event appears but the forecast does not, f
represents the number of occurrences when the forecast event appears but the observed event does not,
and c represents the number of occurrences when neither the forecast nor the observed event appears.

**4.2 Evaluation of different models**

To investigate the differences between the CNN-BiLSTM-AM model and traditional machine learning
algorithms, this study compared their forecasting performance on convective weather test dataset from
2015 to 2017. Figure 5 presents a comparative analysis of predicted and observed precipitation between
the observations and various models. The figure suggested that the SVM and KNN methods resulted in
subpar precipitation prediction, as evidenced by their relatively high RMSE values of 2.42mm and
2.79mm respectively. Compared to the WRF model, these figures represent reductions of 22.68% and
10.86%. Additionally, the correlation coefficients between the predicted and actual precipitation were
low due to more dispersed distributions. On the contrary, RF and GBDT methods demonstrated superior
performance in precipitation prediction. They achieved smaller RMSE values between the predicted and
actual rainfall, reaching 1.62mm and 1.89mm respectively, representing reductions of 48.24% and 39.62%
when compared with the WRF model. These methods also exhibited stronger correlations, indicated by
higher correlation coefficients, suggesting concentrated distributions of predicted rainfall and actual
precipitation errors. However, despite the promising results obtained by RF and GBDT methods, the
CNN-BiLSTM-AM model proposed in this study outperformed them. The distribution of the predicted
and actual precipitation using the CNN-BiLSTM-AM method was notably more concentrated, leading
to the smallest overall error. Specifically, the RMSE value for the CNN-BiLSTM-AM model was merely
1.22mm, marking reductions of 61.02%, 49.59%, 56.27%, 35.45%, and 24.9% in comparison to WRF,
SVM, KNN, GBDT, and RF models respectively. Furthermore, it reached an impressive correlation
coefficient of approximately 99% between the predicted precipitation and the actual data. Based on this
evidence, it is clear that the CNN-BiLSTM-AM model substantially surpasses traditional machine-



learning algorithms such as SVM, KNN, GBDT, and RF in the context of precipitation prediction.

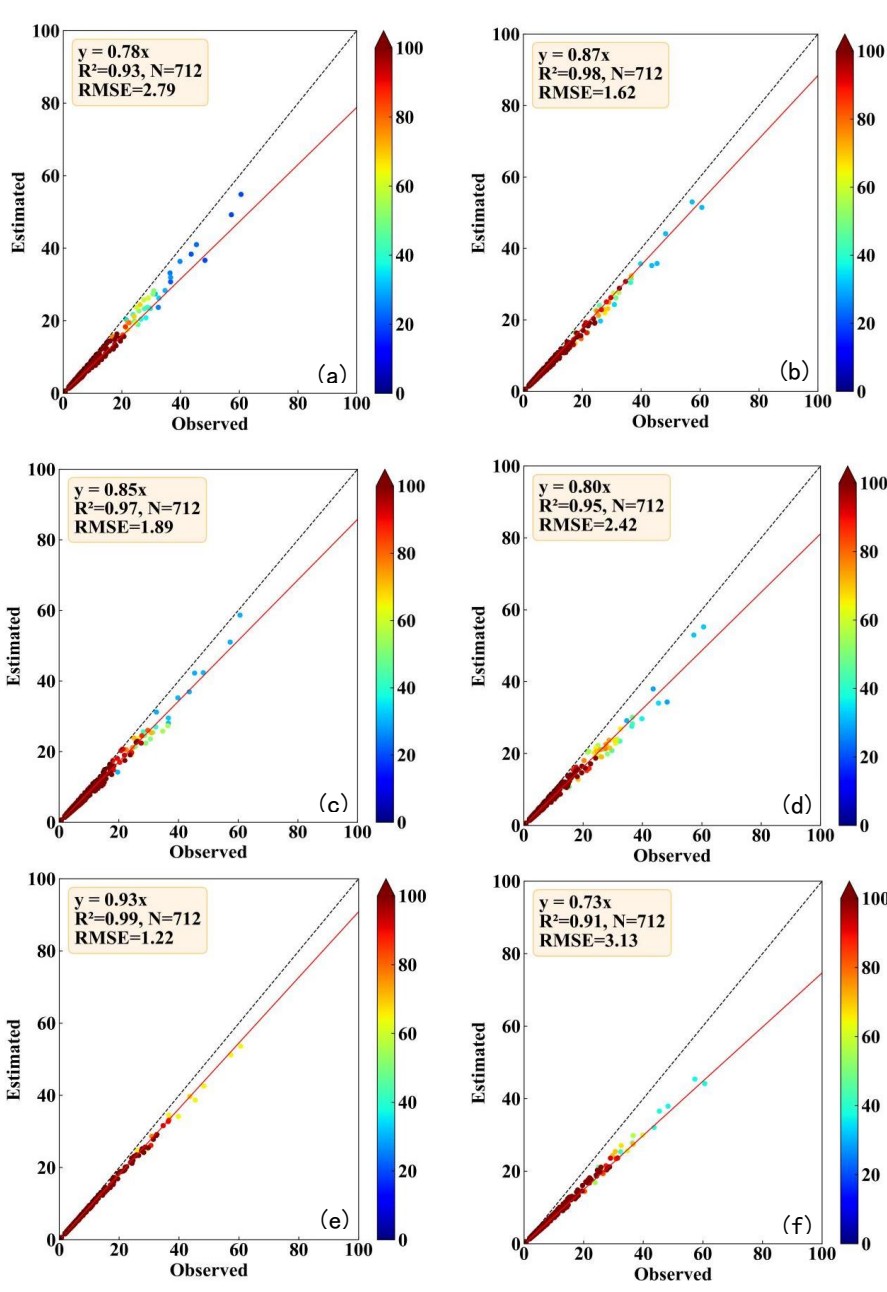


**Figure 5: Scattered density plots of the observed and machine-learning corrected precipitation (a: 10-fold**
**cross-validation training dataset of KNN model, b: 10-fold cross-validation training dataset of RF model, c:**
**10-fold cross-validation training dataset of GBDT model, d: 10-fold cross-validation training dataset of SVM**
**model, e: CNN-BiLSTM-AM model forecasts, and f: WRF forecasts)**



To further investigate the reliability of the CNN-BiLSTM-AM model, we compared it with the other five
models using the cumulative distribution probability scatter plots and Taylor plots, and the results were
shown in Figure 6. From the data provided in the figure, it was evident that the CNN-BiLSTM-AM
model outperforms the other models significantly. It exhibited a standard deviation of 1.02 and a
correlation coefficient of 0.99. Following closely behind as the second most accurate model is the RF
model, boasting a standard deviation of 1.12 and a correlation coefficient of 0.97. The KNN model
demonstrated the weakest performance, with a standard deviation and correlation coefficient of 1.26 and
0.90 respectively. The accuracy metrics of the SVM, and GBDT models fell between those of the
superior (CNN-BiLSTM-AM and RF) and inferior (KNN) models. Specifically, their standard deviations
and correlation coefficients were recorded as follows: SVM - 1.24 and 0.91; GBDT - 1.20 and 0.94. In
conclusion, the CNN-BiLSTM-AM model held a distinct advantage in its ability to effectively extract
the development characteristics of convective weather, thereby achieving superior precipitation
prediction.

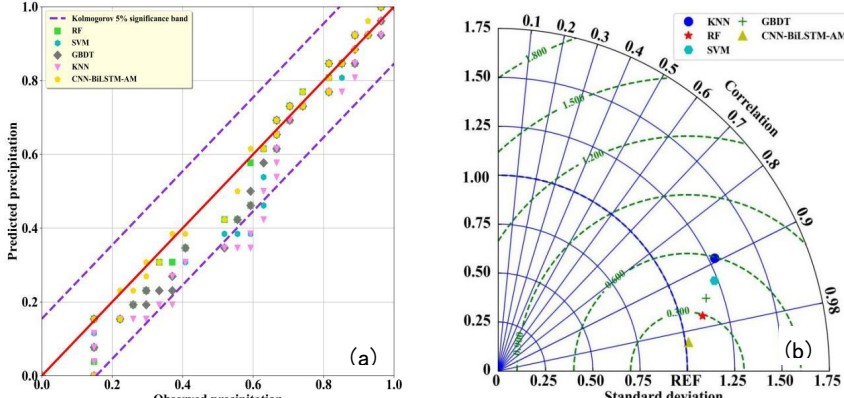


**Figure 6: The cumulative distribution probability scatter plots of the observed precipitation and the predicted**
**precipitation of 6 models(a) ; Taylor distribution plot of different model performance(b)**
**4.3 Individual Case Forecast Evaluation**
On July 22, 2022, widespread thunderstorms and short-term heavy precipitation occurred in Henan and
central Inner Mongolia, China. The forecast performance of this weather process is illustrated in Table
2 and Figure 7. From the comparison of various prediction methods in the table, we can conclude that:
Regardless of whether it was the CNN-BiLSTM-AM model or other algorithms such as Gradient



Boosting Decision Trees (GBDT), Random Forests (RF), Support Vector Machines (SVM), or K-Nearest
Neighbors (KNN), the forecast performance for convective weather showed a decreasing trend with
increasing forecast duration. Additionally, the Probability of Detection (POD) and Threat Score (TS)
significantly decreased, while the False Alarm Ratio (FAR) significantly increased. This can be attributed
to the nonlinear and complex nature of convective system development, which leads to diminishing
predictability with longer forecast durations. Among the five learning models tested, the CNN-BiLSTM-
AM model demonstrated the best predictive performance in the 2-6 hour forecast duration. For example,
within the 2-6 hour time range, the POD, Equitable Threat Score (ETS), and TS values for the CNN-
BiLSTM-AM model were 0.550, 0.524, 0.484, 0.470, 0.463, 0.495, 0.453, 0.384, 0.310, 0.245, and 0.440,
0.375, 0.338, 0.306, 0.226, respectively. Compared to the second-ranked RF model, they improved by
15.30%, 23.88%, 17.48%, 16.34%, 19.64%, 21.32%, 26.89%, 21.52%, 14.81%, 20.69%, and 17.33%,
23.76%, 25.19%, 60.21%, 88.33%. Overall, the CNN-BiLSTM-AM model showed a significant
improvement in forecast performance compared to other machine learning algorithms. As the forecast
duration increased, the CNN-BiLSTM-AM model outperformed other machine learning models in terms
of forecast results. For example, at the 3rd hour, the difference in TS values between the CNN-BiLSTM-
AM model and GBDT, RF, SVM, KNN models were 0.129, 0.072, 0.152, 0.163, representing
improvements of 52.44%, 23.76%, 68.16%, 76.89%. At the 6th hour, the corresponding TS differences
were 0.110, 0.106, 0.111, 0.112, showing improvements of 94.83%, 88.33%, 96.52%, 98.25%. Therefore,
it can be observed that beyond the 2-hour forecast stage, the CNN-BiLSTM-AM model performs more
prominently in convective weather forecasting. The figure 6 clearly illustrates the variations in forecast
performance across different models. In the 1-hour forecast duration, both deep learning and machine
learning algorithms did not exhibit significant changes in forecast performance. This can be attributed to
the fact that within the 0-1 hour time range, convective system morphology undergoes minimal changes,
and the NCEP fnl analysis data contains sufficient information about the initial stage of convection,
allowing effective prediction relying solely on analysis data. However, within the 2-6 hour forecast
duration, the CNN-BiLSTM-AM model consistently outperformed the machine learning methods,
particularly as the lead time extended. The performance gap between the CNN-BiLSTM-AM model and
other machine learning models gradually widened. This indicates that the CNN-BiLSTM-AM model
possesses nonlinear evolution recognition and prediction capabilities, with its provided information



becoming more valuable as the forecast duration increases. These findings further confirm the
advantages of deep learning methods in severe convection forecasting.
**Table 2. Comparison of various models on convective weather case on July 22, 2022**

| Models | Forecast duration | POD | FAR | ETS | TS |
|---|---|---|---|---|---|
| CNN-BiLSTM-AM | 1h | 0.618 | 0.389 | 0.501 | 0.484 |
| | 2h | 0.550 | 0.430 | 0.495 | 0.440 |
| | 3h | 0.524 | 0.454 | 0.453 | 0.375 |
| | 4h | 0.484 | 0.482 | 0.384 | 0.338 |
| | 5h | 0.470 | 0.538 | 0.310 | 0.306 |
| | 6h | 0.463 | 0.646 | 0.245 | 0.226 |
| GBDT | 1h | 0.633 | 0.386 | 0.486 | 0.473 |
| | 2h | 0.433 | 0.558 | 0.277 | 0.319 |
| | 3h | 0.427 | 0.652 | 0.200 | 0.246 |
| | 4h | 0.409 | 0.733 | 0.145 | 0.197 |
| | 5h | 0.402 | 0.783 | 0.111 | 0.164 |
| | 6h | 0.372 | 0.824 | 0.083 | 0.116 |
| RF | 1h | 0.651 | 0.412 | 0.477 | 0.481 |
| | 2h | 0.477 | 0.525 | 0.408 | 0.375 |
| | 3h | 0.423 | 0.632 | 0.357 | 0.303 |
| | 4h | 0.412 | 0.721 | 0.316 | 0.270 |
| | 5h | 0.404 | 0.790 | 0.270 | 0.191 |
| | 6h | 0.387 | 0.803 | 0.203 | 0.120 |
| SVM | 1h | 0.413 | 0.405 | 0.459 | 0.451 |
| | 2h | 0.249 | 0.545 | 0.138 | 0.271 |
| | 3h | 0.186 | 0.636 | 0.088 | 0.223 |
| | 4h | 0.126 | 0.710 | 0.047 | 0.182 |
| | 5h | 0.094 | 0.790 | 0.025 | 0.170 |
| | 6h | 0.076 | 0.854 | 0.013 | 0.115 |
| KNN | 1h | 0.391 | 0.400 | 0.328 | 0.253 |





| | | | | |
|---|---|---|---|---|
| 2h | 0.332 | 0.512 | 0.165 | 0.443 |
| 3h | 0.281 | 0.616 | 0.143 | 0.212 |
| 4h | 0.245 | 0.709 | 0.098 | 0.180 |
| 5h | 0.170 | 0.801 | 0.062 | 0.142 |
| 6h | 0.101 | 0.868 | 0.038 | 0.114 |

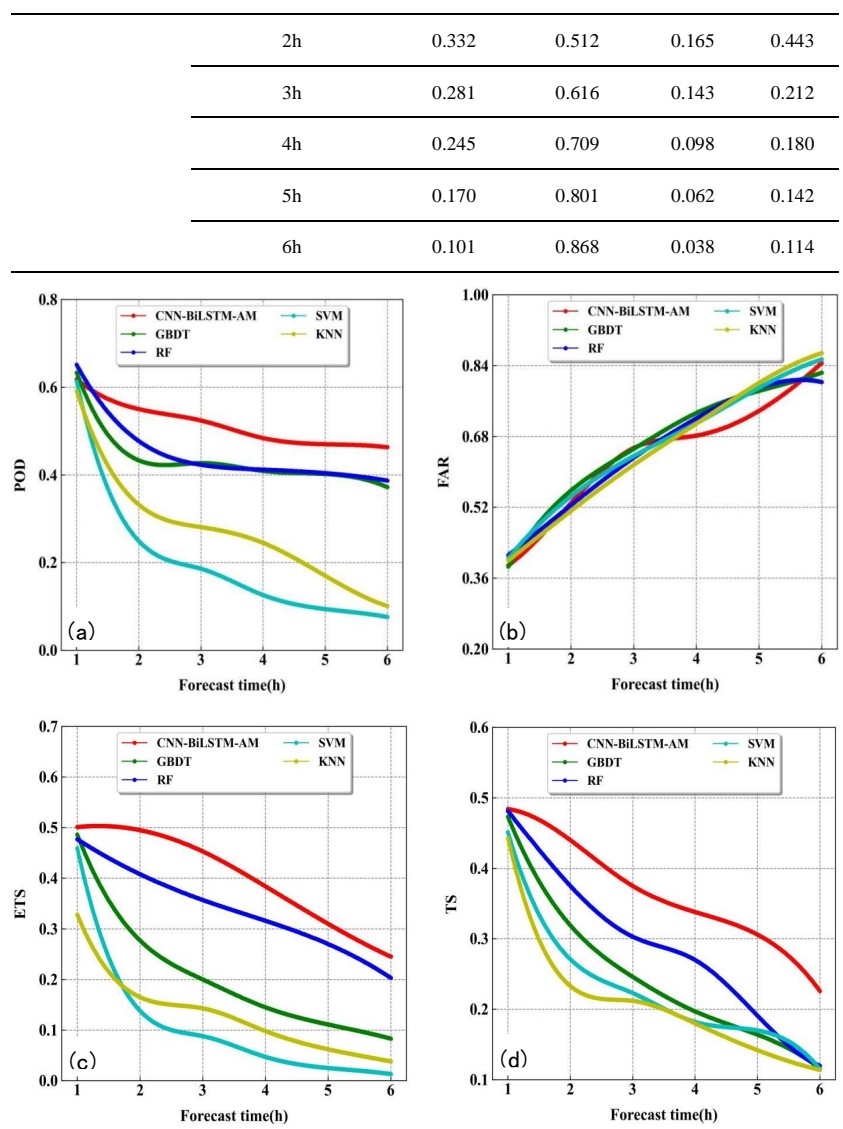


**Figure 7: Comparison of forecast performance of different models**
To assess how errors vary in different models, we selected Zhengzhou in Henan as specific region for
evaluation and analysis. The boxplots of the predicted precipitation and the actual precipitation of 6
models at 12 stations in Zhengzhou show that(see Figure 8), the CNN-BiLSTM-AM gave more accurate
results than the other models, its difference between the observed precipitation and the predicted
precipitation was very small, which is significantly superior to those of the other models; For the RF and
GBDT models, the difference between the observed precipitation and the predicted precipitation was not



significant and both showed better performance than KNN and SVM models. Overall, the CNN-
BiLSTM-AM model showed the best performance with higher accuracy for all stations, and the KNN
and SVM model illustrated the lowest performance among other models and approaches.

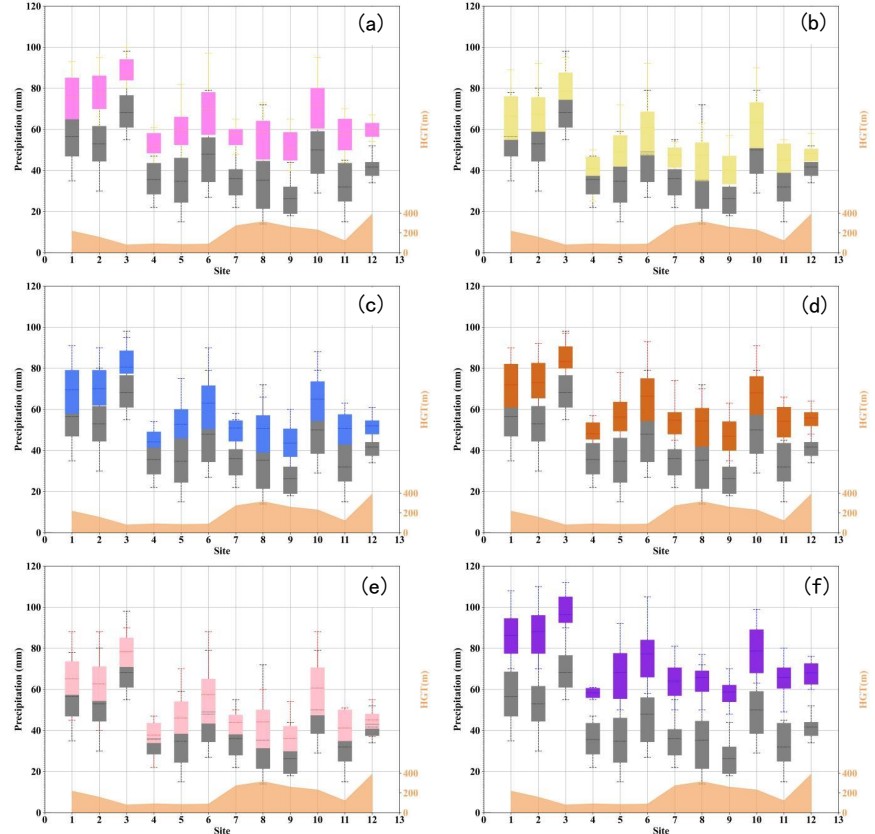


**Figure 8: The boxplots of the predicted precipitation of the KNN(a), RF(b), GBDT(c), SVM(d), CNN-BiLSTM-AM(e), and WRF(f) models at 12stations and the boxplots of the actual precipitation (gray).**

**4.4 Spatial–temporal variations in the best model**
Figure 9 depicts the diurnal variations offered by diverse models in July 2022, and also shows the diurnal
fluctuation in precipitation in the initial WRF forecast. The precipitation forecast by the original WRF
weather prediction model exhibits noticeable inaccuracies. As can be seen from the figure, WRF's
precipitation forecast displays a distinct diurnal variation trait, characterized by substantial discrepancies
between early morning and afternoon hours, namely between 9:00 am and 13:00 pm (Figure 9f). This





indicates that WRF's precipitation forecast tends to be inaccurate and displays significant errors in diurnal
variation.
Compared to the precipitation forecast results of different ML models, the diurnal variation error was
considerably lessened (Figure 9a, b, c, d, and e). Initially, the mean precipitation forecast by the CNN-
BiLSTM-AM model aligns well with the actual average precipitation trajectory, with minimal error and
devoid of diurnal variation (Figure 9e). This suggests that the predicted and actual distributions of
precipitation are in agreement. However, the results performed between 9:00 am and 13:00 pm during
January 2021 were not very satisfactory. This could be attributed to the inadequate generalization
abilities of the training model and the excessive volatility of actual precipitation at these specific times.
Based on the above comparative analysis, it can be inferred that the CNN-BiLSTM-AM model
outperforms other models.
In oreder to facilitate a more intuitive comparison, we visualized the distribution of forecast results for
different models (see Figure 10). As can be seen from Figure 10, the RMSE (Figure 10a, b, c, d, and e)
distribution of precipitation of 5 models show that the performance of the CNN-BiLSTM-AM model is
better than the other machine learning models, RMSE value is mostly between 0.11mm and 3.87mm.
The performances of the RF, SVM, GBDT, and KNN models are not as good as CNN-BiLSTM-AM,
their RMSE were recorded as follows: RF: 0.10mm-4.25mm, SVM: 0.38mm-4.31mm; GBDT: 0.33mm-
4.68mm, KNN: 0.56mm-4.82mm. From the 24-hour forecast scores (Fig.8f), the CNN-BiLSTM-AM
model consistently outperformed the subjective predictions of the forecasters. This indicates that the
CNN-BiLSTM-AM model, based on deep learning techniques, significantly improved the forecast
accuracy for this severe convective weather event, providing forecasters with valuable guidance and
reference.

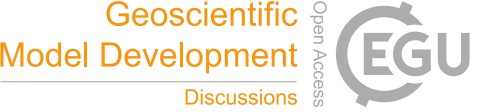

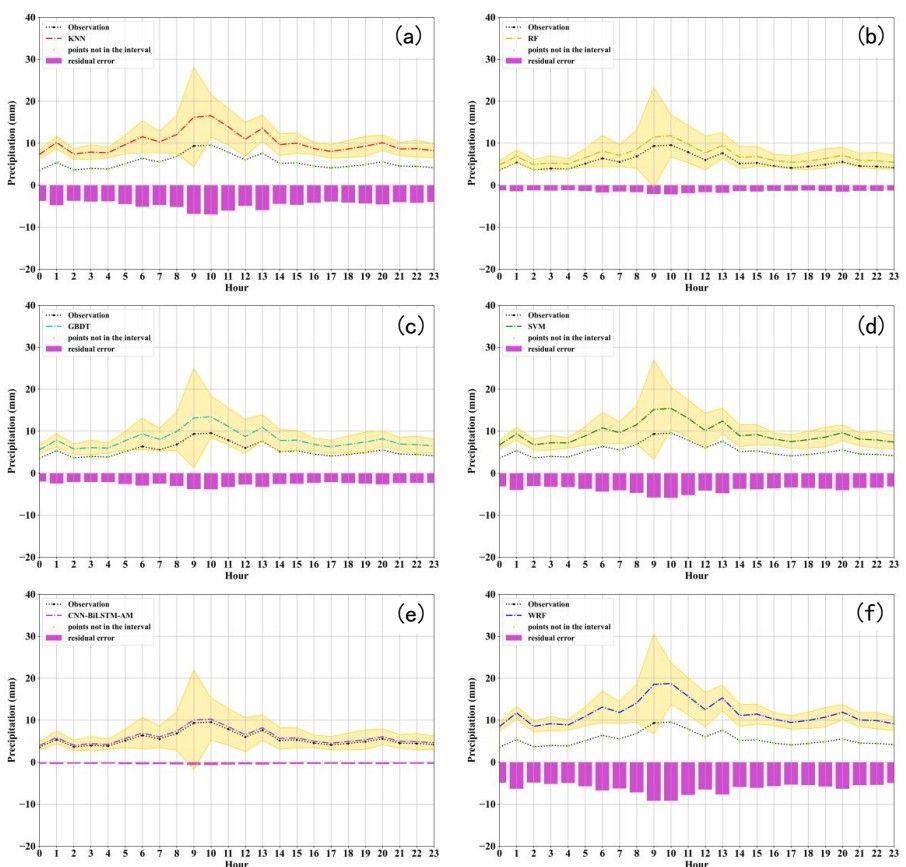

**Figure 9: KNN(a), RF(b), GBDT(c), SVM(d), CNN-BiLSTM-AM(e), and WRF(f) daily variation of predicted**
**and actual precipitation on July 2022.**





**Figure 10: RMSE distribution of CNN-BiLSTM-AM(a), RF(b), SVM(c), KNN(d), GBDT(e) models in Henan;**

**TS scores of strong convective weather on July 22, 2022 (f)**





**5 Discussion**

**5.1 Stability analysis of the proposed models**

The preceding results presented the visualized outcomes of various correction methods, which may not fully prove the stability of different approaches. To further evaluate the stability of various machine-learning models, we compared their performance on strong convective weather forecasting during the flood season (April-September) from 2020 to 2022 by using six evaluation metrics: RMSE, FAR, MAR, POD, TS, and accuracy. The specific results are presented in Figure 11. This comprehensive analysis will provide a more accurate assessment of the stability of different models.

The RMSE values of the five machine-learning models were lower than that of the WRF model, with the CNN-BiLSTM-AM model having the smallest RMSE of 1.12mm. This represented a 63.04% reduction in RMSE compared to the output precipitation of the WRF model. From the perspective of solving regression problems, the model effectively corrected the deviation in precipitation predicted by the numerical forecast model. In comparison to other machine-learning algorithms, the accuracy of the CNN-BiLSTM-AM model showed a significant improvement, demonstrating the stability of deep-learning methods for nonlinear problems such as precipitation, often achieving superior application results.

FAR and MAR are two important indicators for evaluating precipitation forecasting accuracy, reflecting false and missing alarm rates, respectively. As shown in the figure, the FAR of the CNN-BiLSTM-AM model was higher than that of all the other corrected models, while its MAR was lower than that of all the other models. This situation might be due to the fact that while the CNN-BiLSTM-AM model effectively fits precipitation, it also has side effects, leading to precipitation forecasts in the absence of actual precipitation. The FAR and MAR values obtained by the other four machine-learning algorithms were lower than those of the WRF model, indicating that these machine-learning methods can reduce the false and missing rates of the WRF model precipitation forecasts to a certain extent.

Finally, the POD,TS and accuracy scores of the five machine-learning models were significantly higher than that of the WRF model, with the CNN-BiLSTM-AM model achieving the best performance among all models. These results indicate that the CNN-BiLSTM-AM model exhibits an ideal performance in correcting precipitation forecasts from the WRF model, outperforming other machine-learning methods in terms of precipitation correction.



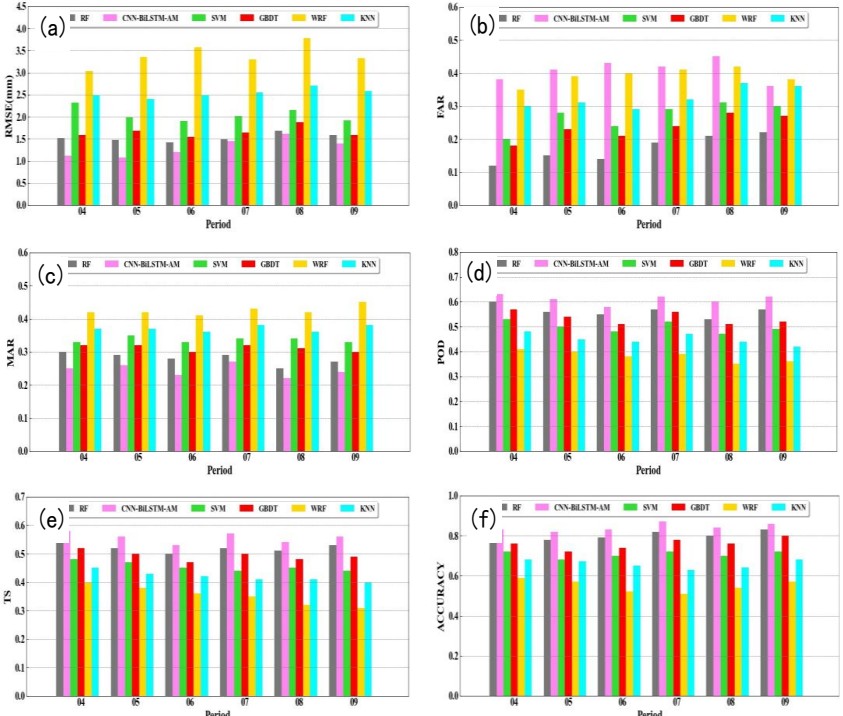


**Figure 11: Evaluation histograms of precipitation predicted by 6 models and actual precipitation in different**
**months ((a), (b), (c), (d), (e), and (f) represent RMSE(mm), FAR, MAR, POD, TS, and ACCURACY**
**respectively).**
**5.2 Explainability of the mechanism of severe convective weather based on machine learning**
Although machine learning and deep learning methods have made significant breakthroughs in various
fields, effectively predicting severe convective weather using these methods still remains a "black box"
challenge (McGovern et al., 2019), with the specific details being difficult to ascertain. Currently, many
researchers are attempting to uncover this "black box" and have developed several model interpretation
and visualization (MIV) techniques. Through MIV, users of machine learning can gain a better
understanding of the strengths, weaknesses, and optimal application scenarios of the models, thereby
increasing trust in the models and enhancing their practicality. If the machine learning forecasts
outperform human forecasters, MIV can also help improve subjective judgment and prediction results,
as well as validate new scientific hypotheses and conjectures (McGovern et al., 2019). In this section,
we apply a method of ranking the importance of forecast factors to analyze the forecasting process of the





deep learning model, aiming to unravel the mystery of the "black box" in deep learning for predicting
severe convective weather.
**5.2.1 Technical method**
By employing the Random Forest (RF) algorithm, we can conduct an importance analysis of forecast
factors to ascertain the significance of each predictor and establish a priority ranking. The fundamental
principle involves measuring the contribution of each feature in every tree within the random forest,
averaging these values, and then comparing the contributions among the features. Typically, we can
utilize the Gini index or Out-of-Bag (OOB) error rate as evaluation metrics. In this study, we solely focus
on the approach that employs the Gini index for assessment. Here, we denote Variable Importance
Measures (VIM) as the score reflecting the importance of variables, GI represents the Gini index,
$VIM_j^{(Gini)}$ denotes the Gini index score for the j-th feature ($X_j$). Assuming there are J features, $X_1$, $X_2$,
$X_3$, ..., $X_J$, I decision trees, and C categories, the Gini index of node q in the i-th tree is calculated as
follows:

$$GI_q^{(i)} = \sum_{c=1}^{|C|} \sum_{c' \neq c} P_{qc}^{(i)} P_{qc'}^{(i)} = 1 - \sum_{c=1}^{|C|} (P_{qc}^{(i)})^2 \qquad (1.9)$$

Among them, C represents the categories, and $P_{qc}$ denotes the proportion of category c at node q. The
change in Gini index for a feature is given by:

$$VIM_j^{(Gini)(i)} = \sum_{q \in Q} VIM_{jq}^{(Gini)(i)} \qquad (1.10)$$

Suppose there are I trees in the Random Forest (RF), then:

$$VIM_j^{(Gini)} = \sum_{i=1}^{I} VIM_j^{(Gini)(i)} \qquad (1.11)$$

Finally, normalization is performed:





$$VIM_j^{(Gini)} = \frac{VIM_j^{(Gini)}}{\sum_{j'=1}^{J} VIM_{j'}^{(Gini)}} \tag{1.12}$$

### 5.2.2 The interpretability of machine learning models


Using the RF algorithm, we conducted an importance analysis of forecast factors on the training and
testing datasets of CNN-BiLSTM-AM model constructed by us, obtaining the relative rankings of each
forecast factor's importance and their corresponding correlation coefficients (see Figures 12). From the
figures, it is evident that moisture conditions are crucial for severe convective weather, with the most
important feature being precipitable water (PWAT), which significantly outweighs the second-ranked
feature. Geographic location also has a significant impact on severe convective weather, with longitude
(LON) and latitude (LAT) ranking second and third among all indicators, respectively. As convective
weather fundamentally arises from temperature variations, when the near-surface air absorbs sufficient
heat and expands, its density decreases, leading to an unstable atmosphere that triggers convective
weather. Therefore, the temperature (T) feature ranks fourth. Strong convection imposes certain
requirements on atmospheric dynamic lifting conditions, with 300hPa vertical motion (W300) ranking
fifth. Atmospheric energy conditions have some influence on severe convective weather but are not the
most important factors, as Bulk Convective Available Potential Energy (BCAPE) and Lifted Index (LI)
rank sixth and seventh, respectively. These features align closely with the importance distribution of
physical quantities related to short-term severe convective weather obtained by Tian et al. (2015) through
statistical analysis. Additionally, these features are consistent with the characteristics of various types of
severe convective weather analyzed by forecasters in terms of moisture, energy, dynamics, and other
conditions (Zeng & Yang, 2020; Zhang et al., 2020; Zhang et al., 2022).These results demonstrate the
high effectiveness of machine learning in automatically extracting features and further confirm the
rationality of constructing the sample dataset.



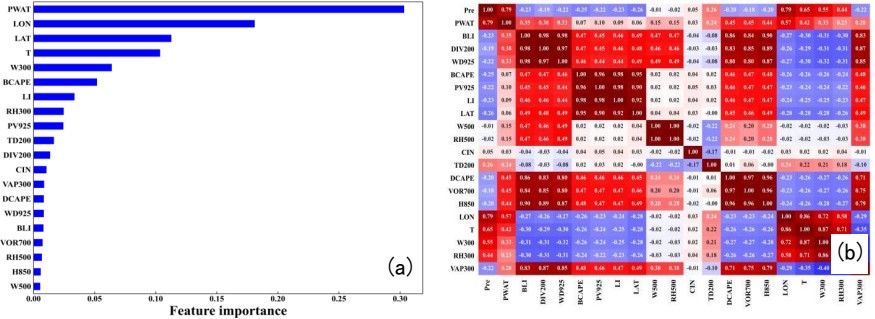

**Figure 12: Relative importance of forecast factors (top 20 most important forecast factors); The correlation coefficient between each of the forecast factors**

## 6 Summary

This study utilizes the NCEP Global Forecast System (GFS) reanalysis data and forecast data to construct a CNN-BiLSTM-AM model for predicting severe convective weather using deep learning techniques and comprehensively evaluate the performance of this model. Furthermore, to gain a better understanding of the "black box" principles of deep learning for severe convective weather prediction, we visualize the training process by ranking the importance of forecast factors. The main conclusions are as follows:

1)Compared to traditional machine learning algorithms such as Gradient Boosting Decision Trees (GBDT), Random Forest (RF), Support Vector Machines (SVM), and K-Nearest Neighbors (KNN), the CNN-BiLSTM-AM model can automatically identify and learn deeper nonlinear features of severe convective weather. Consequently, it achieves higher prediction accuracy on the severe convective weather dataset. Moreover, as the lead time increases, all algorithms exhibit a decreasing trend in their forecasting performance.

2)In comparison to subjective forecasts made by meteorologists, the CNN-BiLSTM-AM model demonstrates good performance in forecasting severe convective weather, with improvements in scoring metrics such as Probability of Detection (POD), False Alarm Rate (FAR), Threat Score (TS), and Equitable Threat Score (ETS). Specifically, the average TS score of the CNN-BiLSTM-AM model for heavy precipitation reaches 0.336, representing a 33.2% improvement compared to the meteorologists' score of 0.252. Additionally, due to its training with a large-scale sample dataset, the model can automatically extract classification features and consider various parameters related to severe convective weather conditions. This enables it to assess the convective conditions at each grid point within the





forecast range, resulting in a lower miss rate than false alarm rate. This lower miss rate provides better
guidance for meteorologists' forecasts.
3)Using the RF algorithm, we perform an importance analysis of forecast factors on the training and
testing datasets of the CNN-BiLSTM-AM model, obtaining the relative rankings and correlation
coefficients of each forecast factor. The analysis results reveal that for severe convective weather,
precipitable water (PWAT) is the most critical moisture condition, with its importance significantly
surpassing the second-ranked feature. Geographic location also has a significant impact, with longitude
(LON) and latitude (LAT) ranking second and third among all factors, respectively. As convective
weather fundamentally arises from temperature variations, temperature (T) ranks fourth. Strong
convection imposes certain requirements on atmospheric dynamic lifting conditions, with 300hPa
vertical motion (W300) ranking fifth. Atmospheric energy conditions have some influence on severe
convective weather but are not the most important factors, as Bulk Convective Available Potential
Energy (BCAPE) and Lifted Index (LI) rank sixth and seventh, respectively. Through this ranking
analysis of forecast factors, we find that the order of importance determined by deep learning for severe
convective weather prediction is roughly consistent with the subjective understanding of meteorologists.
This further demonstrates the effectiveness of deep learning in automatically extracting features for
severe convective weather and verifies the rationality of constructing the sample dataset.



**Code availability**

The code and model are available at Zenodo via https://doi.org/ 10.5281/zenodo.8417134 (Zhang et al., 2023).

**Data Availability**

The data are available at Zenodo via https://doi.org/ 10.5281/zenodo.8417134 (Zhang et al., 2023).

**Author contributions**

YY and ZG were responsible for conceptualization, supervision and funding acquisition. SZ developed the software and prepared the original draft. SZ and YY developed the methodology and carried out formal analysis. XX and SZ validated data. ZG,YY, XX, ZD, and YL were reviewed and edited the text. SZ was responsible for visualization. All authors have read and agreed to the published version of the paper.

**Competing interests**

The authors declare that they have no conflict of interest.

**Financial support**

This research has been supported by the second batch of service public bidding projects for EHV transmission companies in 2022 (2022-FW-2-ZB) (grant no. CG0100022001526556).



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
