# Peer review of "A deep learning method for convective weather forecasting : CNN-BiLSTM-AM (version 1.0)"

_Geoscientific Model Development, 2023_

## Referee Comment (RC1)

Zhang et al. introduced the CNN-BiLSTM-AM model for convective weather prediction in China's Henan region. Their findings indicate that CNN-BiLSTM-AM outperforms traditional machine learning models like Random Forests (RF), Support Vector Machines (SVM), and K-Nearest Neighbors (KNN), as well as a physical model like the Weather Research & Forecasting Model (WRF). Additionally, the authors employed the RF algorithm to assess input importance in the CNN-BiLSTM-AM model, finding that the resulting rankings align with meteorologists' subjective understanding.

I have a few major comments on the manuscript as follows:

1. The current structure and grammar of the manuscript make it very hard to follow:
    a. Why are sections 3: Deep learning models (it should be plural) and 4.1: Evaluation method outside of Section 2: Data and method?
    b. The manuscript should be revised by a native speaker to rectify grammar and phrasing errors. For example, the first sentence of the abstract has no meaning: (i) it's either "we developed" or "this work presents," (ii) the second part of the sentence is not relevant.
2. The Introduction section is incomplete:
    a. The authors should cite the recent Google's GraphCast model (https://www.science.org/doi/10.1126/science.adi2336) as well.
    b. Based on my humble understanding, "forecaster" is a human who makes forecasts about different convective storms, right? If so, what tools does the "forecaster" use for forecasting? I couldn't understand the context of the first paragraph.
    c. Physical-based models like NCEP GFS and WRF need discussion alongside deep learning approaches.
3. The current Section 2 is incomplete:
    a. The authors compared CNN-BiLSTM-AM with other ML models and the WRF model. At the very least, those models should briefly be mentioned in Section 2 as well.
    b. The authors should provide the websites or links to all the datasets for reproducible purposes.
    c. The comma sign in Table 1 is not commonly used.
    d. Lines 163-205 and Figures 3 and 4 are too trivial. The sentences are just repeating from the figures. However, information about the forecasting timestep and the training loss is missing.
    e. Line 227: In my experience, 30 epochs for training is very few. Why did the authors not train more? Was the loss converged?
    f. The training data is available in a 6-hour timestep; how could the authors configure the CNN-BiLSTM-AM so that it can predict every hour?
4. The Section 4 is incomplete:
    a. Figure 8: What are the shaded orange graphs at the bottom?
    b. Figures 8, 9, and 10: the text inside the plot is very small.
    c. Figure 10: What are the spatial resolutions of the predictions?

d. Figure 10: Where and how did you get the results from the human-forecast?
5. I was confused about the whole approach of the manuscript. First, comparing a deep learning model (CNN-BiLSTM-AM) with traditional machine learning models is not a fair comparison. Why didn't the author compare their approaches with current deep learning models that they referenced, such as ConvLSTM or Pangu-Weather? Second, results suggest CNN-BiLSTM-AM has better performances than RF. Why did the authors use a less effective model to explore the importance ranking of inputs for the better model?

Based on these serious flaws, especially the last point, I would recommend rejecting the manuscript.

---

## Community Comment (CC1)

**Reviewer 1**

Zhang et al. introduced the CNN-BiLSTM-AM model for convecve weather prediction in China's Henan region. Their findings indicate that CNN-BiLSTM-AM outperforms traditional machine learning models like Random Forests (RF), Support Vector Machines (SVM), and K-Nearest Neighbors (KNN), as well as a physical model like the Weather Research & Forecasting Model (WRF). Additionally, the authors employed the RF algorithm to assess input importance in the CNN-BiLSTM-AM model, finding that the resulting rankings align with meteorologists' subjective understanding. I have a few major comments on the manuscript as follows:

**Response:** We gratefully thank you for taking the time to provide constructive comments and helpful suggestions for our manuscript, which have significantly raised the quality of the manuscript and enabled us to improve the manuscript. Each suggested revision and comment was accurately incorporated and considered. Below the comments are response point by point.

**Comment 1:** The current structure and grammar of the manuscript make it very hard to follow: a. Why are sections 3: Deep learning models (it should be plural) and 4.1: Evaluation method outside of Section 2: Data and method? b. The manuscript should be revised by a native speaker to rectify grammar and phrasing errors. For example, the first sentence of the abstract has no meaning: (i) it's either "we developed" or "this work presents," (ii) the second part of the sentence is not relevant.

**Response:** Thank you very much for your detailed comments.

**With regard to comment 1a,** we have restructured sections 3 and 4.1 to sections 2:Data and methods.
**With regard to comment 1b,** we will review the entire paper carefully for spelling, punctuation, and grammar errors, and used spell checking tools to read each sentence aloud to identify errors and avoid excessive repetition of words and phrases. Meanwhile, maintain consistency in tense, style, and format throughout the entire paper, checking for consistency in capitalization, phrase matching, title style, and citation format. Finally, use writing aids such as grammar checkers and editing software to identify errors and improve overall writing quality.

**Comment 2:** The Introduction section is incomplete: a. The authors should cite the recent Google's GraphCast model (https://www.science.org/doi/10.1126/science.adi2336) as well. b. Based on my humble understanding, "forecaster" is a human who makes forecasts about different convective storms, right? If so, what tools does the "forecaster" use for forecasting? I couldn't understand the context of the first paragraph. c. Physical-based models like NCEP GFS and WRF need discussion

alongside deep learning approaches.

**Response:** Thank you very much for your detailed comments.

**With regard to comment 2a,** we have consulted and cited the recent Google's GraphCast model in this article, and the following are some of the citations: "GraphCast," a machine learning-based approach trained using reanalysis data, provides predictions for numerous weather variables globally at a resolution of $0.25°$ over a period of 10 days. These predictions can be generated in less than one minute. In terms of accuracy, GraphCast outperforms the most precise operational deterministic systems in 90% of the 1380 verification targets, thereby enhancing the ability to predict SCW events (Remi et al., 2023).

**With regard to comment 2b and 4d,** the human-forecaster here mainly refers to the staff engaged in forecasting business in the meteorological station, because I am a forecaster myself, and I usually need to carry out refined forecasting for multiple stations, and the platforms and jobs used mainly include WRF, MICAPS4.0 and other forecasting models. The reason why we want to introduce the human-forecasters here is to understand how much the CNN-BiLSTM-AM model proposed in this study will outperform human-forecasters, and what effects it will achieve if it is applied and implemented in the future.

**With regard to comment 2c,** we have added NCEP GFS and WRF discussion in section 1, here are some excerpts: "Many researchers use numerical weather prediction (NWP) models, such as the Weather Research and Forecasting (WRF) model, the Global Forecast System (GFS), as valuable tools for predicting SCW (Quenum et al., 2022; Varlas et al., 2021). This is especially crucial for early warning purposes (Giannaros et al., 2022). However, due to the inherent instability of the atmosphere, current numerical atmospheric models struggle to accurately predict the specific occurrence time, location, and intensity of SCW processes. In a study conducted by Kryza et al. (2013), the WRF model was utilized to forecast short-term heavy precipitation in southwestern Poland. The results demonstrated that none of the model configurations were able to accurately reproduce local heavy precipitation. To address this limitation, Hamill et al. (2012) applied an ensemble forecasting system to enhance the WRF's precipitation forecasting ability. While ensemble forecasting can partially reflect the forecasting ability or reliability of the real atmosphere, it cannot improve the physical mechanisms of models. The Global Forecast System (GFS) is one of the most widely used global weather forecast models, providing predictions for weather conditions around the world. The GFS model ingests a vast amount of observational data from various sources, including satellites, weather stations, radars, and buoys, to initialize its initial conditions. Nonetheless, the hydrostatic spectral dynamical core of the GFS (Sela, 1980) has not undergone substantial upgrades since the 1980s, despite improvements in spatial resolution, energy conservation, and computational efficiency (Juang, 2004, 2008; Eckermann, 2009; Yang, 2009). Although a global nonhydrostatic spectral model is theoretically feasible (Juang, 1992), its poor scalability makes it impractical for future computing architectures.

Consequently, extending the GFS spectral dynamical core to nonhydrostatic scales for predicting convective-scale events, which require a grid spacing of less than 4 km, is generally not considered a viable solution (Weisman et al., 1997; Done et al., 2004; Roberts and Lean, 2008; Prein et al., 2015). While NWP serves as an important means of SCW prediction by simulating atmospheric processes through mathematical and physical equations, there are many uncertainties in the numerical prediction process, such as inaccurate initial conditions and parameterization of physical processes. Due to the chaotic nature of the atmosphere, these inaccuracies can result in significant uncertainties in model results (Stevens et al., 2013). Over the past two decades, advancements in observation technology, data assimilation techniques, model resolution, physical parameterization, and statistical post-processing of model output have effectively improved the prediction results of numerical weather models. However, there are still numerous challenges and uncertainties that need to be addressed within the model system components."

**Comment 3:** The current Section 2 is incomplete: a. The authors compared CNN-BiLSTM-AM with other ML models and the WRF model. At the very least, those models should briefly be mentioned in Section 2 as well. b. The authors should provide the websites or links to all the datasets for reproducible purposes. c. The comma sign in Table 1 is not commonly used. d. Lines 163-205 and Figures 3 and 4 are too trivial. The sentences are just repeating from the figures. However, information about the forecasting timestep and the training loss is missing. e. Line 227: In my experience, 30 epochs for training is very few. Why did the authors not train more? Was the loss converged? f. The training data is available in a 6-hour timestep; how could the authors configure the CNN-BiLSTM-AM so that it can predict every hour?
.
**Response:** Thank you very much for your detailed comments.

**With regard to comment 3a and comment 5: "comparing a deep learning model (CNN-BiLSTM-AM) with traditional machine learning models is not a fair comparison",** we will introduce the current well-established deep learning network models such as ConvLSTM, Predrnn++, CNN, FC-LSTM, LSTM, to compare with the CNN-BiLSTM-AM model, and re-conduct the experiments and analysis.
**With regard to comment 3b,** we have provided the websites or links to all the datasets for reproducible purposes. Among them, ERA5-Land hourly data were derived from https://cds.climate. copernicus.eu. The severe convective weather observations were obtained from https://data.cma.cn/.
**With regard to comment 3c,** we have made revisions to the comma sign in Table 1.
**With regard to comment 3d,** we have removed the redundant part and added the training loss, here are some excerpts: The training process of CNN-BiLSTM-AM involves the following steps: 1) Input Data: Provide necessary data for training. 2) Input Data Normalization: Standardize input data to address significant variations using Equation (1.1). 3) Network Initialization: Initialize weights and biases in each

layer. 4) CNN Layer Computation: Sequentially pass input data through convolutional and pooling layers to extract features and obtain output values. 5) BiLSTM Layer Computation: Use BiLSTM hidden layer to compute output data from the CNN layer. 6) AM Layer Computation: Compute output data from BiLSTM layer using the AM layer. 7) Output Layer Computation: Calculate model output value based on AM layer's output value. 8) Error Calculation: Compare computed output with true values and calculate error. 9) Check Termination Conditions: Determine if termination conditions are met, such as completing cycles or reaching weight or prediction error thresholds. If met, training is completed; otherwise, continue training. 10) Error Backpropagation: Propagate errors in opposite direction, update weights and biases, and return to step 4 for continued training.

$$y_i = \frac{x_i - \bar{x}}{s} \tag{1.1}$$

Among them, $y_i$ is the standardized value, $x_i$ is the input data, $\bar{x}$ is the average value of the input data, and s is the standard deviation of the input data.

The prediction process of CNN-BiLSTM-AM consists of the following main steps: 1) Input Data: Provide the input data required for prediction. 2) Data Standardization: Normalize the input data. 3) Prediction: Feed the standardized data into the trained CNN-BiLSTM-AM model and obtain the corresponding output values. 4) Data Standardization Recovery: The output values obtained from CNN-BiLSTM-AM are in standardized form. To restore them to their original values, apply Equation (1.2) to convert the standardized values back. 5) Output Results: Present the recovered results after restoration as the completion of the prediction process.

$$x_i = y_i * s + \bar{x} \tag{1.2}$$

Among them, $x_i$ represents the recovered value of the standardized value, $y_i$ represents the output value of CNN-BiLSTM-AM, s represents the standard deviation of the input data, and $\bar{x}$ represents the mean value of the input data.

During the training process, we utilized the ADAM optimizer (Kingma and Ba, 2014) with a learning rate set at 10-4, and default values were used for other settings (Perol et al., 2017). The CNN-BiLSTM-AM model can incorporate ERA5-Land hourly data and transform features into a four-dimensional array of $M \times 28 \times 32 \times 1$ (M represents the number of samples), enabling predictions for SCW. The model training process employed the early stop strategy, with an iteration period (Epoch) set to 300. If the loss did not decrease for more than 10 epochs, the operation was automatically terminated. A batch size of 16 was used. The loss function selected for minimization during training was mean squared error (MSE). The formula is as follows:

$$MSE = \frac{1}{m} \sum_{i=1}^{m} (y_i' - y_i)^2 \tag{1.3}$$

Where m represents the training sample size, $y_i$ represents the actual value, $y_i'$ represents the predicted value. Statistical measures, including correlation coefficient

(r), standard deviation ($\sigma_n$), and root-mean-square error (RMSE) were used to assess model performance.

**With regard to comment 3e,** we have made adjustments to Epoch, the model training process employed the early stop strategy, with an iteration period (Epoch) set to 300.

**With regard to comment 3f,** we will utilize a subset of ECMWF's ERA5 archive to train and evaluate CNN-BiLSTM-AM. ERA5 is the fifth-generation reanalysis data generated by ECMWF using the Integrated Forecasting System (IFS cycle 41r2), served as the source for our datasets. These datasets provide comprehensive information on the global atmospheric changes since 1940. In comparison to ERA-Interim, ERA5 incorporates updates in the atmospheric model and assimilation system while also assimilating a larger volume of observational data (Hersbach et al., 2020). As the most current reanalysis product from ECMWF, ERA5 boasts a spatial resolution of 0.25° degrees along with a temporal resolution of 1 hour.

**Comment 4:** The Section 4 is incomplete: a. Figure 8: What are the shaded orange graphs at the bottom? b. Figures 8, 9, and 10: the text inside the plot is very small. c. Figure 10: What are the spatial resolutions of the predictions? d. Figure 10: Where and how did you get the results from the human-forecast?

**Response:** Thank you very much for your detailed comments.

**With regard to comment 4a,** the shaded orange graphs at the bottom represents the altitude of the 12 stations.

**With regard to comment 4b,** after all the experiments have been performed, we will redraw Figures 8,9,and 10 based on the latest experimental results.

**With regard to comment 4c,** we will add a scale bar in Figure 10 to facilitate the understanding of resolution in the following time.

**With regard to comment 4d,** same reason as comment 2b.

**Comment 5:** I was confused about the whole approach of the manuscript. First, comparing a deep learning model (CNN-BiLSTM-AM) with traditional machine learning models is not a fair comparison. Why didn't the author compare their approaches with current deep learning models that they referenced, such as ConvLSTM or Pangu-Weather? Second, results suggest CNN-BiLSTM-AM has better performances than RF. Why did the authors use a less effective model to explore the importance ranking of inputs for the better model?

**Response:** Thank you very much for your detailed comments.

Firstly, we will introduce the current well-established deep learning network models such as ConvLSTM, Predrnn++, CNN, FC-LSTM, LSTM, to compare with the CNN-BiLSTM-AM model, and re-conduct the experiments and analysis.

Secondly, the reason for why we use RF to explore the importance ranking of inputs for the better model is that: (1)RF is a commonly employed technique for feature selection. It operates by assessing the importance of each feature, ranking them based on their calculated importance, and subsequently filtering out the most significant ones. This is particularly valuable in scenarios where a substantial number of features are involved in classification or regression tasks. It is common for many features to exhibit high correlation and dimensionality issues. Incorporating these features into the model can have a significant impact on the accuracy of model training and prediction (Breiman, 2001; Robin et al., 2010; McGovern et al., 2019). (2)Although the performance of RF in this study is different from that of CNN-BiLSTM-AM, the gap is very small, and the purpose of further approximating the performance of CNN-BiLSTM-AM can be achieved by optimizing the parameters in the later stage.

**Based on these serious flaws, especially the last point, I would recommend rejecting the manuscript.**

**Response:** In this study, we propose a deep learning model based on ERA5-Land hourly data and observational data, which can be used to achieve refined prediction of severe convective weather, and is of great practical significance and value. Especially in the context of global warming, the risk of climate change is getting higher and higher, the frequency of severe convective weather events is increasing, which will have a serious impact and losses on society, economy and people's lives. However, due to the complexity and variability of the atmospheric system, the limitations of observation methods, and theoretical understanding, the prediction of severe convective weather is still a difficult problem, the technical means and prediction skills that have been put into business application are often unable to accurately and finely predict severe convective weather, or even if there is a prediction, the specific location and precipitation intensity of the prediction often deviate greatly from the actual situation. The CNN-BiLSTM-AM model proposed in this paper can effectively integrate multi-source data, which can accurately predict nonlinear precipitation that is difficult to predict by traditional methods, and has high operation efficiency and convenient deployment. The CNN-BiLSTM-AM model can provide effective support for future refined forecasting, business application and weather research.

Although there are some deficiencies and flaws in the article that need to be revised vigorously, we will definitely make serious revisions in the following time. I believe that with the guidance and help of the experts, the work will make significant progress, and please give us an opportunity, we will cherish the opportunity, revise the article seriously, and put our research results into business practice as soon as possible.

---

## Community Comment (CC2)

**Reviewer 2**

In this work, author use deep learning algorithms to develop a framework including CNN, BiLSTM, and AM for convective weather forecasting, called CNN-BiLTS-AM method. The approach is novel. The NCEP FNLS analysis datasets were exploited as inputs in which several factors related to meteorology, convective physical quantities, and geographical variables were selected by experts. In addition, station measurements were collected and used for model developments and validation purpose. The training and testing datasets were from 2015 – 2020 in which data of randomly selected 72 days use for testing and the remained data used for training. In order to evaluate the proposed method's performance, different machine learning methods such as KNN, RF, GBDT, SVM, and a physical model (WRF) were implemented, and results were compared in testing dataset and an individual case. The comparison results showed that the CNN-BiLSTM-AM overperformed other algorithms in the test cases.However, I have major comments as follows:

**Response:** We gratefully thank you for taking the time to provide constructive comments and helpful suggestions for our manuscript, which have significantly raised the quality of the manuscript and enabled us to improve the manuscript. Each suggested revision and comment was accurately incorporated and considered. Below the comments are response point by point.

**Comment 1:** The problem statement is not clear. Convection weather forecasting is related to many events. As I understand, evaluation is mainly focused on forecasting of precipitation rate (section 4.2, 4.4, 5.1) and precipitation occurrences (section 4.3). In fact, it is two different problems: classification and regression. In this work, please state clearly what is output?

**Response:** Thank you very much for your detailed comments.

By using the ERA5 hourly data and observation data, our study employs DL algorithms to establish a forecasting model for severe convective weather (SCW) called CNN-BiLSTM-AM. The model is capable of generating 0-6 hour short-term precipitation prediction, which is a regression problem. In the following time, we will carefully review the full text and correct any ambiguities.

**Comment 2:** It is also not clear that author develop one, two, or many models to solve the given problems and how to build models. There is no problem with a general deep learning framework as presented in section 3.1. However, loss functions, which are very important in DL, depend on output designs and number of models. How many models were developed and how were corresponding outputs and loss functions in this work?

**Response:** Thank you very much for your detailed comments.

The loss function selected for minimization during training was mean squared error (MSE). The formula is as follows:

$$MSE = \frac{1}{m}\sum_{i=1}^{m}(y_i' - y_i)^2 \tag{1.3}$$

Where m represents the training sample size, $y_i$ represents the actual value, $y_i'$ represents the predicted value. Statistical measures, including correlation coefficient (r), standard deviation ($\sigma_n$), and root-mean-square error (RMSE) were used to assess model performance.

Next, we will introduce the current well-established deep learning network models such as ConvLSTM, Predrnn++, CNN, FC-LSTM, LSTM, to compare with the CNN-BiLSTM-AM model, and re-conduct the experiments and analysis. The loss functions of these models will be elaborated in subsequent research work.

**Comment 3:** There is also inconsistence in construction of training and testing datasets which is key points for deep learning/machine learning algorithms. In section 3.2, authors only mentioned to classification problem. In addition, it is not clear what is used as ground truths to label a grid as positive/negative sample. If all 2400 station measurements in China were collected, the model should be built for all China. However, in experimental area is Henan region (section 2.3) with only 12 stations. Were authors use station precipitation measurements to label positive/negative samples? How many stations were used and its location and distribution? Please report also detail number of training and testing samples for this case? (line 223-225)

**Response:** Thank you very much for your detailed comments.

In this study, observation data were used to label the predictors. These observation data were collected from 2,400 ground stations across the country, but in this study, in order to improve operational efficiency, we only selected the observation data of 119 stations related to the experimental area of Henan for the construction of the database.

**Comment 4:** The dividing of training and testing datasets are not suitable for this problem. Forecasting is for future, so the use of one- or two-year data for testing is more independent. For example, data from 2015 – 2020 can be divided into training dataset (2015 – 2019) and testing dataset (2020). The current division based on randomly selection can bring very good results on modelling but not correct for future and independent datasets.

**Response:** Thank you very much for your detailed comments.

We will reconstruct the training and testing datasets using real-time data from January to December between 2015 and 2020, as well as ERA5 hourly data, with 2015-2019 as the training dataset and 2020 as the testing dataset , and then re-experiment and analyze.

**Comment 5:** In section 3.2, the questions mentioned above are also rising for training and testing datasets for regression problem (precipitation estimation). Author needs to add this information to this section.

**Response:** Thank you very much for your detailed comments.

The model is capable of generating 0-6 hour short-term precipitation prediction, which is a regression problem. In the following time, we will carefully review the section and correct any ambiguities.

**Comment 6:** In section 4.1, many evaluation indexes for regression were not defined. Example: R2, RMSE.

**Response:** Thank you very much for your detailed comments.

We will add the missing evaluation indexes as soon as possible.

**Comment 7:** In section 4.2, the test dataset is from 2015 – 2017 (line 243). It is different from the description of training/testing datasets in Section 3.2 (2015 – 2020, line 222). Also, please point out the predicted parameter is hourly (or daily) precipitation and forecast duration. The observation data is collected from how many ground stations and where they are.

**Response:** Thank you very much for your detailed comments.

In order to maintain the consistency of the article, we will unify the training dataset and the testing dataset in this section. The prediction parameter is hourly precipitation for the next 0-6 hours. Since the experimental area of this study was selected in Henan, China, the observation data used were from 119 ground stations in Henan.We will add the above information timely.

**Comment 8:** In section 4.2, in Figure 8, the boxplots were used to present predicted precipitation. However, this description cannot point out the outperformance of the proposed method and RF because they are likely to each other. Moreover, there is confliction of results in Figure 8 and Figure 5. In Figure 5, the model is underestimated precipitation while in Figure 8, the model is overestimated precipitation. Please explain why.

**Response:** Thank you very much for your detailed comments.

Figure 5 presents a comparative analysis of predicted and observed precipitation on convective weather test dataset from 2015 to 2017. Figure 8 presents a comparative analysis of predicted and observed precipitation at 12 stations in Zhengzhou on July 22, 2022.The difference between the two is that the time scale is different, the object of the analysis is different, and the results obtained are different. From the above results, it can be seen that the performance of the model in long-term forecast timeliness is different from that in short-term forecasts.

**Comment 9:** In figure 9, it is not clear that diurnal variation is calculated at one station, or averages on 12 stations in Henan. If the dataset is the same as in Figure 8, there is disagreement on results of boxplot and line chart which present performance of CNN-BiLSTM-AM. In Figure 9, error of the proposed method is almost equal to 0 while in figure 8, there is difference of prediction and observation presented by boxplot. Please explain why.

**Response:** Thank you very much for your detailed comments.

Figure 8 presents a comparative analysis of predicted and observed precipitation at 12 stations. Figure 9 depicts the diurnal variations offered by diverse models in July 2022. The main difference between the two is the time scale. Figure 8 depicts the distribution of total precipitation at 12 stations over the process of the event on July 22, 2022, while Figure 9 depicts hourly variation over the process, so the results of the analysis will be different.

**Comment 10:** In Figure 10, the RF (b) should have lower error than CNN-BiLSTM-AM (a) based on color scales, and it is conflict with above conclusions. Following this concern, why RF is not used for stability analysis in Section 5.1.

**Response:** Thank you very much for your detailed comments.

In Figure 10, RMSE value of the CNN-BiLSTM-AM model is mostly between 0.11mm and 3.87mm. RMSE value of the RF model is mostly between 0.10mm and 4.25mm. Judging by the results, the performance of the CNN-BiLSTM-AM model is better than that of RF, and is not conflict with above conclusions. Secondly, we have used RF for stability analysis in Section 5.1.